

# Effects of different drying methods on smears of canine blood and effusion fluid

Fiamma G. De Witte[1], Aimee Hebrard[2], Carolyn N. Grimes[3], Kristin Owens[4], Deanna M. Schaefer[2], Xiaojuan Zhu[2] and Michael M. Fry[2]

[1] BluePearl Veterinary Hospital, Levittown, PA, United States of America
[2] Department of Biomedical and Diagnostic Sciences, College of Veterinary Medicine, University of Tennessee, Knoxville, TN, United States of America
[3] Ethos Diagnostics Science, San Diego, CA, United States of America
[4] Antech Diagnostics, Levittown, PA, United States of America

## ABSTRACT

**Background**. Glass slide preparations from a variety of specimens (blood, masses, effusions) are commonly made as part of the diagnostic work-up, however the effects of various drying methods in veterinary practice and diagnostic laboratory settings is not clear.

**Objective**. Compare the effects of four drying methods on results of microscopic examination of canine blood smears and direct smears of pleural or peritoneal effusion fluid.

**Methods**. Twelve canine blood samples (6 from healthy dogs, 6 from sick dogs) and 6 canine peritoneal or pleural effusion samples. Four smears were prepared from each of the 18 samples and dried using the following methods: air-dry, hair dryer with or without heat, and heat block at 58 °C. Observers, blinded to the drying method, independently reviewed the slides microscopically, using a scoring system to evaluate cell morphology and (for blood smears) echinocyte numbers; scoring results were analyzed statistically.

**Results**. For blood smears, several comparisons showed more adverse effects on morphology using the heat block method than for one or more other drying methods. For effusion fluid smears, RBCs dried with the heat block or air-dry methods had more poorly preserved morphology than RBCs dried by the hair dryer method without heat.

**Conclusions and clinical relevance**. The results (1) indicate that different drying methods had a significant effect, (2) support using a hair dryer without heat for both blood smears and effusion fluid smears, and (3) discourage using a 58 °C heat block.

Corresponding author
Fiamma G. De Witte,
fiammagdw@gmail.com

## INTRODUCTION

Veterinary practitioners commonly make glass slide preparations from a variety of specimens (blood, masses, effusions) as part of the diagnostic work-up. Different methods exist to dry blood, effusion fluid, or other tissue samples on glass slides prior to staining, from simple air-drying to methods using an electrical device such as a hair dryer, a fan, or

a heat block. The authors of the present study are all laboratory professionals—4 clinical pathologists board-certified by the American College of Veterinary Pathologists, one of them also a medical technologist licensed by the American Society for Clinical Pathology (ASCP), and another ASCP-licensed medical technologist—and none of us are aware of an established protocol for slide drying. In our experience, air-drying at room temperature is the standard method. The prevalence of various drying methods in veterinary practice and diagnostic laboratory settings is not clear. Anecdotally, opinions vary about the pros and cons of different methods and about whether electrically-assisted drying damages the cells and adversely affects smear interpretation. One veterinary cytology textbook suggests using a hair dryer on low heat setting, or a small fan, but discourages heat fixation because of possible adverse effects on cell morphology (*Meyer, 2016*). A 2006 study of specimens from dogs with ceruminous otitis externa compared numbers of keratinocytes, yeast, bacteria, and neutrophils on slides, with or without heat fixation after air-drying, using two rapid-staining protocols (*Toma et al., 2006*). In that study, heat fixation involved holding a lighter flame under the slide for a few seconds. The authors noted that there was debate about the value of heat fixation, with dermatologists and clinical pathologists being for and against it, respectively. That study found no significant differences in the numbers of those cells or organisms but did find significant differences between the two observers. To our knowledge, there are no published reports of controlled study of the effects of different drying methods on other types of cytology specimens or blood smears.

The objective of the present study was to compare the effects of four drying methods—air-drying at room temperature, use of a hair dryer with heat or without heat, and use of a heat block—on results of microscopic examination of canine blood smears and direct smears of pleural or peritoneal effusion fluid. The null hypothesis was that using a hair dryer or heat block does not introduce any detectable artifact compared to air-drying.

## MATERIAL AND METHODS

Sample recruitment and slide preparation were performed at the University of Tennessee Veterinary Medical Center (UTVMC). A total of 12 blood samples were included, using the first sample submitted for a CBC to the UTVMC Clinical Pathology Laboratory, from two patient groups: 6 samples from dogs presenting to the Community Practice service for an annual or initial patient examination, and 6 samples from dogs presenting to the Small Animal Internal Medicine or Emergency and Critical Care service because of illness. The dogs that presented to Community Practice were considered generally healthy, although some abnormal physical examination or laboratory findings were detected in all of them (the list included nuclear sclerosis, cataracts, dental/periodontal disease, presumptive sebaceous adenoma, dermatitis, osteoarthritis, muscle wasting, and various laboratory abnormalities). Additionally, the study included the first 6 canine peritoneal or pleural effusion samples submitted to the laboratory during the recruitment period. These samples were left over from the routine diagnostic caseload and used in accordance with the UTVMC patient admission procedures and publicly stated policy (*Anonymous, 2020*) that laboratory specimens submitted as part of a patient's diagnostic work-up can be used for

[1]Electric hair dryer: Conair Mid-size Dryer, 1875 watt.

[2]Heating block dryer: Lab-Line Temp-Block Module Heater H2025-5.

research and test development. All samples were obtained during a 12-week period in September to December, 2018.

For each blood or fluid sample, direct smears were prepared and dried sequentially, using constant standardized materials and methods for smear preparation and standardized procedures for each drying method. The same person prepared and dried each sample so that there was consistency from sample to sample. An electric hair dryer[1] with different temperature settings was purchased from a major retailer. The heat block[2] was maintained at 58 °C. Four drying methods were used on smears prepared from each sample (blood and effusion fluid):

- Method 1: Standard (air-drying at room temperature)
- Method 2: Hair dryer—high, regular setting (with heat)
- Method 3: Hair dryer—high, without heat ("cool shot")
- Method 4: Heat block

Each drying method was applied until the smear was visibly dry by gross examination. The order of the drying methods was rotated with each sample (i.e., starting with Method 1 for the first smear sample, Method 2 for the second smear sample, etc.). The distance between the hair dryer and the glass slide was kept constant at 6 inches for both hair dryer methods. Smears were all stained with the same automated aqueous-based Romanowsky-type stain[3], and coverslipped. Initially, smears were labeled to identify the blood sample (patient ID and date) and the drying method. Subsequently, smears were relabeled to enable the slide reviewers to know from which sample the slide was prepared but to remain blinded to the drying method and all patient information besides species, until after all the slide reviews were completed.

[3]Romanowsky-type stain: Wescor Aerospray Aqueous Stainer 7120, Custom Stain #7, Logan, UT.

The 5 authors evaluated the smears, independently and blinded to the drying method and patient information. Reviewers were instructed to assign scores based on the monolayer area of the smear most suitable for detailed morphologic evaluation, using 50x to 100x objective lens magnification. Additionally, reviewers were instructed to scan the entire smear at low magnification (4x objective lens), review the feathered edge of each smear using at least 10x objective lens magnification, and write down any subjective observations about differences between smears that were prepared from the same samples but that were treated differently, even if those differences are not reflected in the scores. Reviewers were not instructed to look for any particular morphologic abnormalities besides echinocytosis. The reviewers rated the cell morphology of RBCs, WBCs, and platelets within the blood smear monolayer and nucleated cells within the smears of effusion fluid (using 50x to 100x objective lens magnification) using a numeric scoring system:

- Score 1: No evidence of introduced artifact from drying method.
- Score 2: Some evidence of abnormal morphology suspected to be an artifact of the drying method, but unlikely to affect diagnostic interpretation (describe the abnormal morphology).

- Score 3: Evidence of abnormal morphology suspected to be an artifact of the drying method, and likely to affect diagnostic interpretation (describe the abnormal morphology).

The RBC echinocytosis scoring on blood smears was based on number of echinocytes observed per 100x objective monolayer field (mean of 10 fields), a modified version of a published system for routine hematology reporting in veterinary laboratories (*Weiss, 1984*):

- Score 1: 10 or fewer
- Score 2: 11–100
- Score 3: 101+

[4]Statistical software: SAS, version 9.4, release TS1M3; MedCalc 18.10.2.

Statistical analysis was performed using commercial software[4] . Inter-rater scoring agreement was analyzed using Cronbach's alpha and intraclass correlation coefficient, and performed both on the complete dataset (5 reviewers, blood and effusion fluid, all cell types) and on combined blood and effusion fluid data for different cell types (RBC, WBC, platelets, and echinocytes). Effects of different drying methods were tested for statistical significance using ANOVA; mean scores from slide reviewers were considered valid for ANOVA if the Cronbach's alpha value was at least 0.5 (*Hinton et al., 2004*). Two-way repeated measures ANOVA was used to analyze mean blood smear scores, with health status ("healthy" or sick) as the between-subject factor, drying method as the within-subject factor, and their interaction. When a low Cronbach's alpha value cast doubt on the validity of the mean scores used for ANOVA and was attributable to a single reviewer's scores being much different from the other four reviewers' scores, then the ANOVA was performed both with and without the discrepant reviewer's scores (i.e., based on a mean of 5 and 4 scores, respectively). One-way repeated measures ANOVA was used to analyze the effect of drying methods on scoring of effusion fluid smears. The least squares means computed and separated with Bonferroni correction methods. Because blood smear WBC scores were right-skewed, the data were transformed using the natural log transformation. The Shapiro–Wilk test and QQ normality plots were used to evaluate normality of ANOVA residuals. A Levene's test was used to assess the equality of variances for the residuals. A $P$ value <0.05 was considered significant.

## RESULTS

A total of 72 slides were available for review: 48 blood smears and 24 direct smears of effusion fluid (4 peritoneal, 2 pleural). One of the blood samples was noted to be grossly lipemic. Raw data for reviewer scoring of all slides are presented as Supplemental Data, along with any subjective observations.

All statistical assumptions regarding normality and equality of variances were met for all analyses. The Cronbach's alpha value was 0.79 among five raters for the complete data set, at least 0.7 for RBC (0.73), platelet (1.0), and echinocytosis (0.9) scoring, and much lower (0.28) for WBC scoring. The lower inter-rater agreement for WBCs was mainly attributable to the scores of one reviewer (one of the clinical pathologists) being noticeably different
**Table 1  Mean scores for blood smear RBCs.** Groups with different superscripts have significantly different scores.

| Health status | Drying method | Mean score (standard deviation) |
|---|---|---|
| Healthy | Air-dry | 1.50[b](0.21) |
| | Hair dryer, without heat | 1.47[b](0.21) |
| | Hair dryer, with heat | 1.50[b](0.21) |
| | Heat block | 2.33[a](0.47) |
| Sick | Air-dry | 1.63[b](0.27) |
| | Hair dryer, without heat | 1.30[b](0.21) |
| | Hair dryer, with heat | 1.30[b](0.21) |
| | Heat block | 1.63[b](0.63) |

**Table 2  Mean scores for blood smear WBCs.** Groups with different superscripts have significantly different scores.

| | Based on 4 reviewers | Based on 5 reviewers |
|---|---|---|
| Drying method | Mean score (standard deviation) | |
| Air-dry | 1.08[ab](0.12) | 1.13[b](0.12) |
| Hair dryer, without heat | 1.04[b](0.10) | 1.22[ab](0.10) |
| Hair dryer, with heat | 1.10[ab](0.17) | 1.32[ab](0.17) |
| Heat block | 1.31[a](0.36) | 1.40[a](0.36) |

from those of the other four reviewers. Omitting the discrepant reviewer, the Cronbach's alpha value for WBC scoring increased to 0.54.

For blood smears, RBC scores (Table 1) had a significant interaction between health status and drying method ($P = 0.02$): smears prepared from samples from dogs that presented to the Community Practice service, and that were dried using the heat block method, had scores significantly different from any other health status-drying method combination ($P < 0.05$). No other significant differences in RBC scoring were detected. For WBCs (Table 2), no interaction between health status and drying method was detected. Only the drying method was a significant variable ($P < 0.05$). Basing the analysis on scoring by all 5 reviewers, scores for smears dried with the heat block method were significantly different from smears dried with the hair dryer without heat method ($P < 0.01$). Basing the analysis on scoring by 4 reviewers, scores for smears dried with the heat block method were significantly worse than for air-dried smears ($P < 0.01$). For platelets, all scores were identical (score $= 1$), so no further analysis was indicated. For echinocytes, no interaction between health status and drying method was detected, and no difference in scores of smears dried by different methods was detected (mean scores were 1.18 to 1.28).

For effusion fluid smears, scores were available for analysis from only four reviewers, because one reviewer's reported scores were not in accordance with the established scoring system. For RBCs (Table 3), drying method was significant ($P < 0.01$): scores for smears dried with the heat block ($P = 0.01$) or air-dry ($P < 0.01$) method were both different from scores for smears dried with the hair dryer without heat method. For WBCs, no

**Table 3  Effusion fluid RBC scoring.** Groups with different superscripts have significantly different scores.

| Drying method | Mean score (standard deviation) |
|---|---|
| Air-dry | 1.50[a] (0.16) |
| Hair dryer, without heat | 1.17[b] (0.13) |
| Hair dryer, with heat | 1.29[ab] (0.25) |
| Heat block | 1.54[a] (0.25) |

difference in scores of smears dried by different methods was detected (mean scores were 1.00 to 1.04); the samples ranged from 0.68 to $13.18 \times 10^3$ nucleated cells per microliter. No platelets were observed in any of the effusion fluid smears, so there were no data to analyze.

## DISCUSSION

The study involved prospectively gathering canine blood and peritoneal or pleural fluid samples, making four smears from each sample, and treating them with different drying protocols. We elected to use those sample types because they are common in clinical practice and because they allowed for greater uniformity of smear preparation than would likely be attainable using samples from solid tissues. We tested four drying methods that we believe are currently in use based on anecdotal information and personal experience: air-drying, which involved the least manipulation and no additional equipment and could be considered a standard method, and three electrically-assisted methods involving drying with a hair dryer or heat block. Five experienced reviewers examined each smear microscopically, independently and blinded to the drying protocol and patient information, using a numeric scoring system to rate morphologic abnormalities suspected to be an artifact of the drying method.

In general, inter-rater agreement using our scoring system was good. We considered mean scores from slide reviewers valid for ANOVA if the Cronbach's alpha value was at least 0.5, based on the suggestion by Hinton et al., that a value of 0.5–0.7 indicates moderate reliability (*Hinton et al., 2004*). Cronbach's alpha is a measure of internal consistency of a test or scale; there is no set threshold for what constitutes an acceptable value, but 0.7 is often considered desirable (*Taber, 2018*; *Tavakol & Dennick, 2011*). Agreement among the reviewers was above that threshold for every category except WBCs. The outlying WBC scores and the aberrant effusion fluid scores were by the same reviewer and occurred because that person interpreted the scoring instructions differently than did the other reviewers. We decided against asking that person to re-score the slides because they were already aware of the results of most of the other reviewers and would no longer be unbiased. Additional statistical analysis showed that drying methods had some significant effects, enabling rejection of the null hypothesis:

- Heat block drying had an adverse effect on blood smear WBC morphology, whether the analysis was based on scoring by 5 reviewers (questionable validity) or 4, and on effusion
fluid smear RBC morphology. We suspect that the finding of a significant interaction between health status and drying method for blood smear RBCs was an example of Type I error (i.e., erroneous rejection of the null hypothesis or false positive), as we have no reason to believe that good general health makes RBCs more susceptible to heat block-induced damage than does illness.

- The hair dryer without heat method yielded better results than the heat block or air-dry method for effusion fluid RBCs, and better results than the heat block method for blood smear WBCs.
- The air-dry method yielded better results than the heat block method for blood smear WBCs but worse results than the hair dryer without heat method for effusion fluid RBCs.
- The hair dryer without heat method tended to produce better results than the hair dryer with heat method, but the differences were not statistically significant.

This study did not show drying method to have a significant effect on echinocyte scoring. We incorporated a blood smear scoring category for echinocytosis because—although echinocytes can occur in association with many pathologic conditions (*Weiss et al., 1990*; *Harvey, 2012*)—they are often considered a drying artifact until proven otherwise (*Stockham, 2008*). Artifactual echinocytes are also known as crenated cells. Echinocytosis scoring was based on average number of abnormal cells per high-power field, consistent with conventional reporting practice (*Weiss, 1984*), but expressing echinocytosis as a percentage of erythrocytes would be a more quantitative method that might yield more meaningful results.

The study design had some limitations. It had low statistical power because of the modest number of samples—we limited enrollment in this initial study to 12 blood samples and 6 effusion samples because microscopic examination and scoring was time-consuming— potentially resulting in Type II error (i.e., failure to detect some significant differences in effects of drying methods). It was designed to test whether using a hair dryer or heat block introduces any detectable artifact compared to air-drying, but not designed to identify or describe any particular type of artifact other than echinocytosis. Only canine samples were included, and only blood and effusion fluid samples were evaluated, so the applicability of the findings to other species and types of samples is not clear. The effusion samples were all of low to moderately increased cellularity, and the applicability of the findings to other types of effusions would also require further study; moreover, many of the cells at the feathered edge of the effusion smears were lysed, irrespective of the drying method, and it is not clear how this might have affected the results. The study did not evaluate potential variability in susceptibility to drying-induced artifacts in individual dogs due to the influence of breed, age, sex, diet, or other factors. The study did not incorporate more than one model of hair dryer, or how varying the drying conditions (time, distance between the hair dryer and the slide) could have affected results. Similarly, we only tested the heat block method under one set of time and temperature conditions. More thorough written instructions, or supplementing the instructions with additional training, might have obviated the problem of low inter-rater agreement for WBCs, and might have resulted in effusion smear scores from all 5 reviewers being available for analysis.

## CONCLUSIONS

To our knowledge, this is the first published report of controlled study of the effects of different drying methods on results of microscopic examination of blood smears and direct smears of pleural or peritoneal effusion fluid. The null hypothesis was that using a hair dryer or heat block does not introduce any detectable artifact compared to air-drying. Despite limitations in sample number and composition, species, and study design, the results enabled rejection of that hypothesis. For blood smears, several comparisons showed more adverse effects on morphology using the heat block method than for one or more other drying methods. For effusion fluid smears, RBCs dried with the heat block or air-dry methods had poorer morphology than RBCs dried by the hair dryer method without heat. Based on the cumulative findings, we recommend use of a hair dryer without heat method for both blood smears and effusion fluid smears, and against the use of a 58 °C heat block. A larger scale study would be required to test the reproducibility of our findings, to more robustly test for differences between drying methods, and to evaluate the effects of different drying methods on other sample types and samples from other species.

**Abbreviations list**

**UTVMC**      University of Tennessee Veterinary Medical Center

## ACKNOWLEDGEMENTS

The authors thank Dr. Bente Flatland for assistance with study design. No third-party funding or support was received in connection with the study design, data analysis, interpretation, writing, or publication of the manuscript.

### Funding

The authors received no funding for this work.

### Competing Interests

Carolyn N. Grimes is employed by Ethos Diagnostic Science and Kristin Owens is employed by Antech Diagnostics. The other authors declare that they have no competing interests.

### Author Contributions

- Fiamma G. De Witte conceived and designed the experiments, analyzed the data, prepared figures and/or tables, authored or reviewed drafts of the paper, and approved the final draft.
- Aimee Hebrard, Carolyn N. Grimes and Kristin Owens performed the experiments, authored or reviewed drafts of the paper, and approved the final draft.
- Deanna M. Schaefer performed the experiments, analyzed the data, authored or reviewed drafts of the paper, and approved the final draft.
- Xiaojuan Zhu analyzed the data, prepared figures and/or tables, authored or reviewed drafts of the paper, and approved the final draft.

- Michael M. Fry conceived and designed the experiments, performed the experiments, analyzed the data, prepared figures and/or tables, authored or reviewed drafts of the paper, and approved the final draft.

## Data Availability

The raw data of cytology scoring are available as a Supplemental File.

## Supplemental Information

Supplemental information for this article can be found online at http://dx.doi.org/10.7717/peerj.10092#supplemental-information.

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
