# Peer review of "Effects of different drying methods on smears of canine blood and effusion fluid"

_PeerJ, doi:10.7717/peerj.10092_

## Round 0.1 · original submission · Major Revisions

Please note the two thorough reviews who have raised concerns that you might be able to address. In addition to those concerns, please consider the following:

The overall study design should be assessment of 4 independent methods for cell preservation rather than partial assessment of independent methods but in other regards consideration of findings relative to "negative" control.

L76: Method 1 is probably not a "negative" control but rather the standard.
L99: Nucleated cells in the smears of effusion fluid were assessed, but Table 3 refers to RBC scoring? What was the cytomorphologic composition and interpretation of the effusion slides?
L102: Should be worded as "no difference from standard method" or similar. All slides were dried, thus this is not an accurate statement. The degree of echinocytosis and other morphologic features was scored in all samples in a blinded manner, and these results should be compared in relation to the nature of slide preparation.
L109: As the authors surely appreciate, fields of monolayers are variably dense, and expressing the number of echinocytes as a percentage of RBC would be more meaningful and probably more reproducible than a relatively subjective score.
L136: As indicated by reviewer 1, information on the health or disease status of the patients is required, and could be provided as a supplement.
L185: Would it be possible for outlying reviewer to review all slides again in accordance with the scoring system?
Table 1: Are these scores for echinocytes only? If so, the title should reflect that.
Table 2: What aspects of WBC were scored? That needs to be detailed in the M&M and alluded to in the table title. Note "in" rather than "on" in column header.
Note comment re: Table 3 above.

Reviewer 1 ·

Basic reporting

The writing was relatively clear and supported understanding of the tables provided. In a few sections the writing was slightly awkward, but not enough to warrant revision.

One major failing is that very few references provided. Only two of the five references were from primary journal articles (and one of these was from 1984). Support is needed for the rationale behind study and discussion of the findings in the report.

The article was structured adequately and orderly, however, another major failing of the article is that a large portion of the discussion was simply restating the results (lines 189 to 204) as opposed to critically discussing them. Proper discussion with possible reasoning behind the positive and negative results (preferably supported by literature) is required. If no literature is available in either human or veterinary medicine to support these results, then this needs to be addressed.

The experiment is self-contained and the results are relevant to the hypothesis.

Experimental design

The article does appear to provide primary research within the aims and scope of this journal.

A failing of the paper is that the rationale behind the research is not well-defined. Lines 45-46 indicate that "anecdotally" there are varying opinions, which is insufficient justification for the research. The outcome of a non-diagnostic or incorrectly interpreted sample should be discussed in the introduction (i.e. cost to the client, possible consequences of incorrect diagnosis) and supported by literature (either human or veterinary articles). Are there any primary articles that support not using heat fixation because of adverse effects on morphology? Are there any standardized procedures in human medicine for drying slides? Further investigation into the literature is required to support the rationale behind this study. If there are few to no articles supporting the rationale, this needs to be explained more clearly.

The "Materials and Methods" section also raises some concerns. The duration/time period of study enrollment should be provided. Addressing the reason for the low sample size may also be useful, as the samples (at least blood samples) should not have been difficult to obtain from a clinical pathology laboratory. There is no information on how the "healthy" dogs were assessed to be healthy (i.e. just physical exam, CBC, biochemistry?). The exact number of pleural vs peritoneal effusions should be mentioned, even if they do not directly impact the results. Line 69 should provide a reference to the website as opposed to a direct link.

Lines 71-73 says the slides were prepared "using constant standardized materials and methods" - for blood smears this is likely adequate, but fluid preparation may not be standardized between labs (i.e. in some labs a cytocentrifuge preparation is standard for fluid samples and others will use sediment smear preparation or line smears) - the exact method of slide preparation for these samples should be noted. Similarly, there is no "routine" staining method with Romanowsky-type stains (lines 83-84) - the exact time in fixation, solution I, and solution II should be noted - if not standardized between slides, this should also be noted.

The remaining methods were straightforward and well-described. The scoring system was described well and information for statistical review was provided.

The raw data table is difficult to understand. Are the scores the “average” score from all 4 or 5 evaluators? Or did each evaluator only get 14-15 slides to review? If the latter is the case, this should be noted in the materials and methods and the evaluator should be identified for each slide (i.e. evaluator 1-5).

The rationale for why the reviewer who incorrectly scored the slides was not asked to review the slides again or replaced with another reviewer should be provided.

Validity of the findings

The study, as described in the paper, appears to be novel and has practical implications (though these implications are not currently described in the paper). Although the results are relatively minor, they would still be a meaningful addition to the literature, especially considering this topic is rarely investigated.

I trust that the statistics were performed correctly, but am not going to investigate them thoroughly (especially given my concerns regarding the raw data). The tables are reasonably straightforward. It is understandable that the WBC table was omitted as it did not show any significant findings.

The conclusions are adequate for the findings of the paper - however, as mentioned above, further elaboration on these results should be addressed in the discussion.

Reviewer 2 ·

Basic reporting

No comment

Experimental design

No comment

Validity of the findings

No comment

Reviewer 3 ·

Basic reporting

I was interested to read this manuscript and have always wondered how the various drying methods and or environmental conditions may interfere with interpretation of blood smears, however I was unaware that a heat block method is still used.
Your style, presentation and English is generally clear, and you hypothesis is clear. Your introduction and literature review is sparse, there is no mention of echinocyte morphology despite scoring of echinocytes, there should be some mention of why these cells are important using journal article references as opposed to textbooks. There may be no published reports on drying methods but there must be reports on the type of artifacts seen in cells and their significance/cause.
The raw data is shared, but needs more explanation. Are the scores an average or mean of the all investigators, are they a cumulative score? it is not specified. Can the raw data be presented in a way that divides the healthy and sick dogs?
The structure of the article and tables is good, but there is no need to present the data from all 5 reviewers in Table 2 when the data was deemed to be unreliable.

Experimental design

The research question is well defined and meaningful, but maybe there should be more explanation in the introduction about why distinguishing artefact from pathology in morphology is so important.
Line 71: were all smears made by the same person (or equally trained and experienced people) to have consistent quality and not introduce another confounding factor that may affect morphology?
Line 80 was there an average time to drying with the methods, as environmental factors can effect drying time (humidity).
Line 87: were the reviewers blinded as to health and illness?
Line 96: I am interested why the reviewers made comments, as presented in the raw data, but that data was not reflected on or appear to be analyzed. What was the purpose? How was it used?
Line 100: I a curious how you modified Weiss's scoring as the journal article did not score artifacts from what I read, or did you use it only for the scoring of the echinocytes. If so the reference should be with the RBC echinocytes not with the artifact scoring.
What was your definition of an artifact, or was that left up to the reviewer?

Validity of the findings

Underlying data were provided, but not all data was used (see comments above). I am concerned about the use of the alpha value of 0.5 as being acceptable (line 121,183), most papers (see Bland and Altman, Tavokol) indicate that anything <0.7 is of questionable value for internal consistency, the minimum recommended is between 0.65-8 with >0.9 being the best result. Using this value the WBC are of questions significance even after eliminating the one reviewer.
Line 159: the raw data for the effusions does not appear complete, as only a morphology score is presented, not raw scores for RBC's or WBC'S, and most comments are low cellularity and 'difficult to evaluate cells'. These comments would make any assessment of the data suspect, maybe data should maybe presented in a descriptive way as any conclusions bases on statistics may not be reliable or representative based on the sample quality.
In the conclusions it is stated that the difference in health and heat block is due to a type 1 error. Do you have any other speculation? There is much mention in the raw data of 'smudge cells', were these found in the healthy dogs due to lipemia? if so, there may be another reason that does relate to health on dogs indirectly, and that might be if all samples were not consistently fasted. This aspect should be considered.

The limitations were clearly outlined, and the conclusions are likely valid for the red cell smears, but are questionable for the effusions smears.

---

## Round 0.2 · Minor Revisions

The additional edits suggested by reviewer 3 should be addressed, and the Table with data on effusions including WBC and RBC should be included as a supplemental table.

Reviewer 1 ·

Basic reporting

no comment

Experimental design

no comment

Validity of the findings

no comment

Additional comments

Most concerns from previous review have been addressed and manuscript appears ok for publication.

Reviewer 3 ·

Basic reporting

no comment, see below

Experimental design

no comment, see below

Validity of the findings

no comment, see below

Additional comments

Thank you for the replies, corrections and additions to the queries raised. The supplemental data is easier to understand with the lay out and the additional data. Most of the reviewers and editor’s comments were addressed. There are still a few that may require some additional comments.

Line 50 There may not be information directly related to blood smears and heat fixation but there are papers that look at the morphology of bacteria, red cells and cell morphology with heat fixation vs methanol fixation from Gram stains (in 1980’s) and more recently papers that looked at ear cytology without and without heat fixation.

Line 106-108 In the methods you requested the evaluators make subjective observations on the smears and the information is presented in the supplementary data. I realize you mention in the discussion how this was in case you wanted to analyze the data at some later point, but since there are instructions to do this and the data is presented, it should be analyzed in some way, or not included in either the methods, supplement or discussion.

Line 112. It would be helpful for you to define what you think is an morphologic abnormality due to or suspected to be artifact of drying, as in the discussion you mention this again with no indication of what this is but leave it up to the evaluator. People differ on what they associate with an artifact of drying vs smears making etc (essentially you have that information in the subjective observations). Not everyone reading this manuscript will be clinical pathologists. some may be student/researchers in other fields making smears but not realizing what artifacts are.

Line 244-, the sentence seems out of place and might be better suited to be modified to fit into the prior section on echinocytes Line 225

---

## Round 0.3 · accepted · Accept

I agree with the authors that the subjective observations should be retained, and I appreciate their additional comment regarding the limitations of such observations.